# Capacity Assessment in Freight-Passengers Complex Railway Nodes: Trieste Case Study

Atieh Kianinejadoshah *[ID] and Stefano Ricci [ID]

Department of Civil, Building and Environmental Engineering, Transport Area,
University of Rome "La Sapienza", Via Eudossiana 18, 00184 Rome, Italy
* Correspondence: atieh.kianinejadoshah@uniroma1.it

**Abstract:** An integrated approach to node and station operation analysis is possible by means of analytical methods, customized to this scope. Alternatively, the simulation models allow more in-depth analyses aiming at the optimization of the use of capacity. The general goals of the research are the comparison of methods for the assessment of railway lines and nodes' capacity, suitability for specific tasks, and stability of the results under variable scenarios. The comparison is finalised to quantify the relative level of confidence of the concerned literature methods. The work is part of a larger research project with the final goal of identifying the most appropriate approach for the optimization of the network capacity and the setup of specific guidelines. In this framework and perspective, the paper introduces synthetically the methods and applies them systematically to a real complex mixed-traffic network in Trieste, situated in Northeast Italy, including the main passengers and freight stations and a set of lines used for both services.

**Keywords:** railway node; capacity; timetabling; simulation; signalling





## 1. Introduction

In the majority of situations worldwide, the increase in railway traffic cannot be matched by an equivalent increase in network capacity, resulting in crescent challenges for maintaining the high performances necessary to play a key in the economic systems of many countries. An increase in the number of train movements can also increase the risk of conflicts among trains [1] and the related potential generation of delays.

Railway capacity is the idealized concept of the maximum number of trains running on a railway system; however, the high rates of use of the concerned railway system make them vulnerable to disturbances, meaning that trains cannot meet their planned timetables due to delays generation and propagation [2,3]. Delays can be due to various causes: disruptions in the operation flow, accidents, malfunctioning or damage equipment, construction work, repair work, and service weather conditions like snow and ice, floods and landslides. Although trains should respect a fixed schedule called nominal timetable, train delays occur daily and can negatively affect rail operation causing service disruptions and economic drawbacks in the worst cases [4]. In passenger and freight services, the operation must be dimensioned on the demand. Therefore, the demand should be the first indicator to plan stops and timetables [5].

An effective rail transit timetable should ensure train punctuality [6]. Therefore a reliable method to increase the quality of railway traffic is to design robust timetables, in which trains can keep their originally planned slots despite small disturbances and without causing unrecoverable delays to other trains. A robust timetable should also be able to recover from small delays. With a robust timetable, railway traffic delays can be reduced, and punctuality can be improved [7]. Timetabling is a well-known optimization problem in railway operations [8] at a tactical level planning. However, the fluctuations in volumes of goods are not fully estimable within the annual timetabling process [9], reason

why the likelihood of delays among freight trains is larger than among passenger trains. Unfortunately, with increasing traffic volume, delay propagation is increasing too. That is the case because the intervals between consecutive trains are smaller for higher traffic volumes, which makes it more likely that one trains' delay impacts other trains' punctuality as well [10].

This paper is presenting developments of previous works, starting from an analytical comparison of the Potthoff method for node and the UIC406 method [11] for line capacity calculation, referred to a common level of punctuality. Since the Potthoff method is considering the random distribution of arrivals [12], the focus is here on the Müller method [13], based on timetable and better comparable, in terms of punctuality results, with traffic simulations. The systematic comparisons of results achievable by OpenTrack® simulation and Müller analytical approach for validation purposes is a key object of the present work.

The outline of the paper is as follows: In Section 2, the history of the research is analysed and compared with the most recent applications of analytical and simulation approaches. The case study of the Trieste real network is described and characterized in Section 3, with the purpose to focus on the cross-validation of the approaches. Computational experiences are further under discussion in Section 4 and some closing remarks are in Section 5.

## 2. State of the Art

In an era where particular attention lays on the use of railway systems, the importance of quantifying network performance, identifying bottlenecks, and defining the margins of railway capacity for operating more trains is relevant.

### 2.1. Line Capacity

The capacity of a railway line is the ability to operate trains with acceptable punctuality [14]. The problem of analyzing the capability of a rail line to absorb additional traffic has become increasingly important recently as various means of consolidating the expanding usage of the rail systems have been proposed [15].

Besides the well-known UIC 405 [16] approach, detailed reviews of several synthetic methods [17] describe that FS Formula does not take into account the heterogeneous traffic. Based on determinations by Canciani [18], treating the capacity of double-track lines with the alternating movement of two classes of speed considering the delay time due to overtaking. According to RFI method [19], treating values drawn up in a medium headway plus margin expansion and extra time. The method defines the theoretical and commercial capacity.

A useful description of some analytical is by Bianchi [20] who applies the queuing theory by analyzing the distribution of intervals between trains and service times, thus providing a contribution to the relationships between capacity and characteristics of track layout and operation of double and single track lines, by taking into consideration the influence of the promiscuous running of trains with reciprocal priority rights in case of crossing and overtaking. The German Federal Railways (DB) have developed a probabilistic method for the quantification of the capacity of railway lines by DB Formula, based on principles similar to the UIC approach, albeit with some significant peculiarities by Vincuna. The UIC 405 R method was officially replaced in 2004 by the compression method UIC 406 [21] as a standard on capacity at the European level. Corriere [22], proposed a method that takes into account the effect of delays in analogy to the road flows. Genovesi [23], proposed an extension of what was considered by Corriere (1982) in order to extend the concept of delay efficiency to stop perturbations and introduced the coefficient of stability in order to obtain more realistic capacity values. Whenever the scheduled timetables for analyzed lines are available, it could be possible to follow the CUI approach for the calculation of capacity. The UIC 406 (2004) capacity method is based on the blocking time theory to measure the capacity occupation of a given timetable, which is achieved by compressing the train blocking time stairways used for assessing practical capacity.

Originally, UIC 406 (2004) described a method for evaluating the capacity of line sections. In the 2nd edition, UIC 406 (2013) expanded the approach to the capacity assessment of nodes.

*2.2. Node Capacity*

Some methods are focused on capacity analysis for railway stations and investigate the relationship between the analytical model proposed by the Müller method and largely applied [24–26] where nodes in the railway network commonly tend to act as bottlenecks limiting the capacity of the entire network. In particular, a review of some capacity methods for complex railway nodes [27] provided by a synthetic approach starting from the Potthoff method [28] to identify the major capacity constraints. He classified the simple junctions showing how the complex junctions can be decomposed and treated as a combination of simple junctions, which could be crossed by the maximum possible average number of trains at the same time. Another useful literature review with a different approach for capacity evaluation of complex railway nodes was presented by Mussone [29].

The optimization model for strategic decision-making in railway station design enables the comparison and selection of a station layout that maximizes the theoretical infrastructure capacity, completely independent of details regarding train sequences and timetables [30].

*2.3. Combined Line-Node Capacity*

All the evolution of research and the continuous increase of interest in railway capacity offer an extraordinarily rich bibliography of existing methodologies developed since the 1950s [31–33] and progressively updated by the most recent comprehensive approach for combined node-lines capacity calculations of complex railway networks [34–36].

Traditionally, the line models require the knowledge of the train succession, and the node models are based on the hypothesis of independence by the surrounding lines. The effect of the interactions among the network elements (lines and nodes), interdependence between carrying capacity, and system characteristics are formalized in [37–40].

According to Ciuffini 2019 [41], the dependence of capacity on the characteristics of the infrastructure is due to the constraints of distance, crossing, and cutting at intersections and stations.

The timetable compression method issued by UIC Code 406 afterward its independent adaptation needs for extension [42,43] applied for the capacity assessment of railway stations and line segments.

*2.4. Simulation*

Simulation was defined by Robinson [44] as "a model that mimics reality" and by Gamerman [45] as "treatment of real problem through reproduction in an environment controlled by the experimenter" used to validate the results of other methods classified as macro-simulation and micro-simulation tools that allow the reproduction of the railway operation based on an extended database [46]. Macro-simulation models use a simplified infrastructure model to reduce computational time and therefore allows simulation of larger network, while micro-simulation offers a description of infrastructure which reproduced the functionality of interlocking, safety and block system. Synchronous micro-simulation tools, such as the commercial software RailSYS® [47], VILLON® [48] and Trenissimo® [49].

In this framework, OPENTRACK® is one of the most accurate and widely applicable simulators to determine the performances of a railway network and to analyze the capacity of lines and stations as well as the robustness of timetables [50].

**3. Materials and Methods**

Despite the recent impacts of mobility restrictions which many countries introduced and motivated by COVID-19, the longer-term statistics show that the share of rail traffic has seen tremendous growth. The growth has been observed in terms of the number of passengers traveling and in terms of freight movements as well [51].

This is the reason why the present research focuses on a complex mixed-traffic network operation and, specifically, the case study of Trieste node, situated in the Northeast of Italy, including the main passengers and freight stations and a set of lines used for both services.

This node lays along the Mediterranean Corridor, central East-West axis in the TEN-T Network. The corridor is 3000 km long and provides a multimodal link for ports of the Western Mediterranean with the centre of the EU. It will let a modal shift from road to rail and connect some of the significant urban areas of the EU [52].

The port of Trieste is located on the Adriatic coast, because of the port's nautical and geographical advantages, traffic is increasing rapidly in terms of both total yearly throughput and railway share for freight transfers Despite the long penetration in the Adriatic Sea, marine traffic is interested in calling at Trieste because it shortens the distance to Central Europe compared to other ports and reduces the transit time compared to North European range ports [53].

In this operational context, the passenger trains departing from Trieste Centrale meet the freight trains from Trieste Campo Marzio station at the Barcola Junction and continue along the double-track line toward Bivio D'Aurisina as Figure 1, and the capacity depends on various parameters, classifiable into two groups:

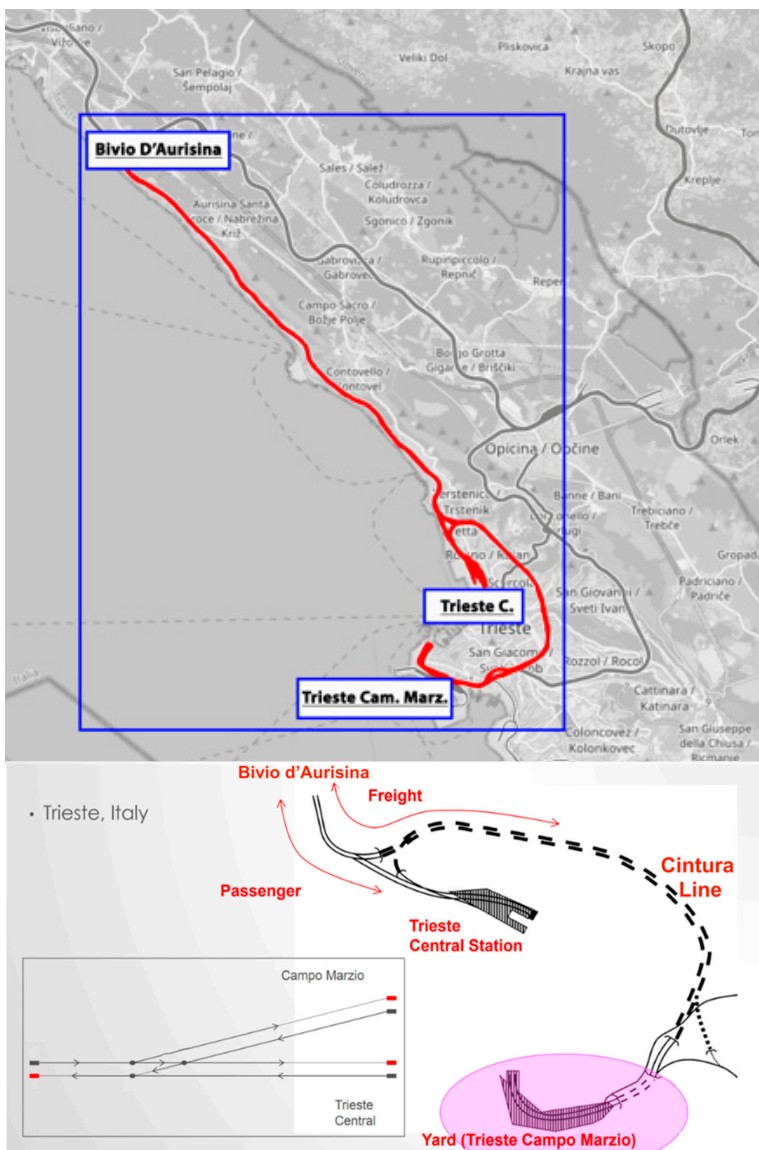

**Figure 1.** Trieste railway node.

Infrastructure parameters:
- Geometry of the tracks (straight, curves, etc.);
- Characteristics of the signaling system (type of block, number and length of sections, minimum distance between trains, braking distance, etc.;
- Layout of stations.

Network effects:
- Operational models (passengers/goods, short/long-distance traffic, etc.);
- Rolling stock (length of trains, acceleration/deceleration performances, etc.);
- Operational time (day, peak periods, etc.);
- Timetabling (running time, buffer time, dwell time, occupation/interdiction time among trains, etc.).

The proposed method aims to offer an innovative contribution to the evaluation of capacity in critical points, namely line-node converging sections, which caused bottlenecks in the system and make it sensitive to delay propagation and capacity reduction.

*3.1. Analytical Method*

The Potthoff method includes combinatorial procedures able to quantify the utilization rate of single routes, station areas and the station as a whole. This method assumes that trains could arrive at any instant with the same probability within the reference time T; therefore, it does not require an assigned timetable, which simplifies its application [54]. The application considered the following hypothesis:
- Passenger trains reference time: 18 h;
- Freight trains reference time: 22 h;
- Passenger traffic in both directions for station Trieste Centrale: 106 trains/day;
- Freight traffic in both directions for station Trieste Campo Marzio: 31 trains/day;
- Intermediate dwell time: 1 min for passenger trains, no stops for freight trains;
- Maximum speed along the line: 60 km/h;
- Maximum speed on deviations: 30 km/h.

With the same input data, the Müller method allows a further step forward by timetabling.

The Müller method is applicable to large railway nodes with the aim of evaluating the potential of a station system using synthetic indicators, measuring their variability and, therefore, the response of the system in the perturbation Ricci [55].

Occupation and interdiction times required by trains and based on a *Poisson* distribution function with:
- Constant density of arrivals probability in *T*;
- Probability to meet a not permissive signal on the line *i*.

The conflicts, depending upon timetable and arrivals, generate probabilistic delays according to process and hypotheses 1 of an equation:

$$R_{ij} = \frac{n_i n_j t_{ij}^2}{2T},$$  (1)

- *R* = medium delay that the trains on the line *i* suffered by waiting on the passing of another train from line *j*;
- *t* = waiting time of the train;
- *n* = number of trains;
- *T* = reference time.

The calculation of the buffer time for delay recovery is possible for any timetable and the following assumptions apply to the present research:
- Traditional automatic block signalling system;
- Overall buffer time is calculated as the average of entry and exit buffers;

- Paths are categorized according to the destinations usable as alternatives for train dispatching;
- Departure time from the origin station is based on the same original timetable for both Müller method and OpenTrack® simulation.

Figure 2 shows the Trieste Centrale (in blue) and Trieste Campo Marzio (in red) average delay/train generated by a variable amount of running trains and the global utilization rates corresponding to them according to the increase of traffic. The yellow circle is showing the present number of trains in both directions for each station.

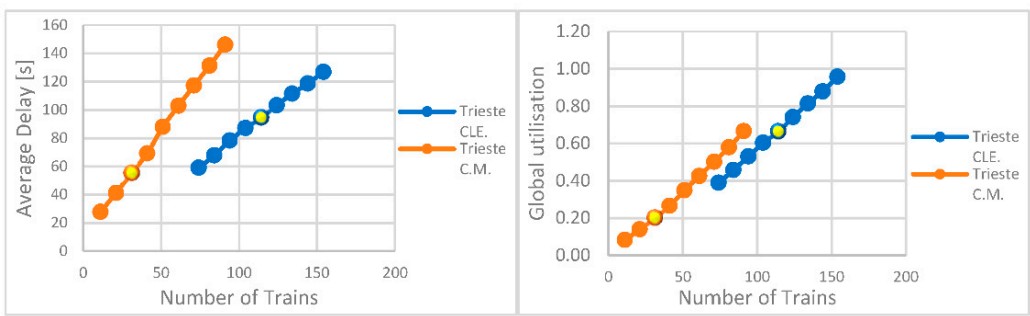

**Figure 2.** Average delay/train and global utilization rates were calculated for stations Trieste Centrale and Trieste Campo Marzio.

Figure 3 shows the architecture of the automatic calculation procedure setup for nodes.

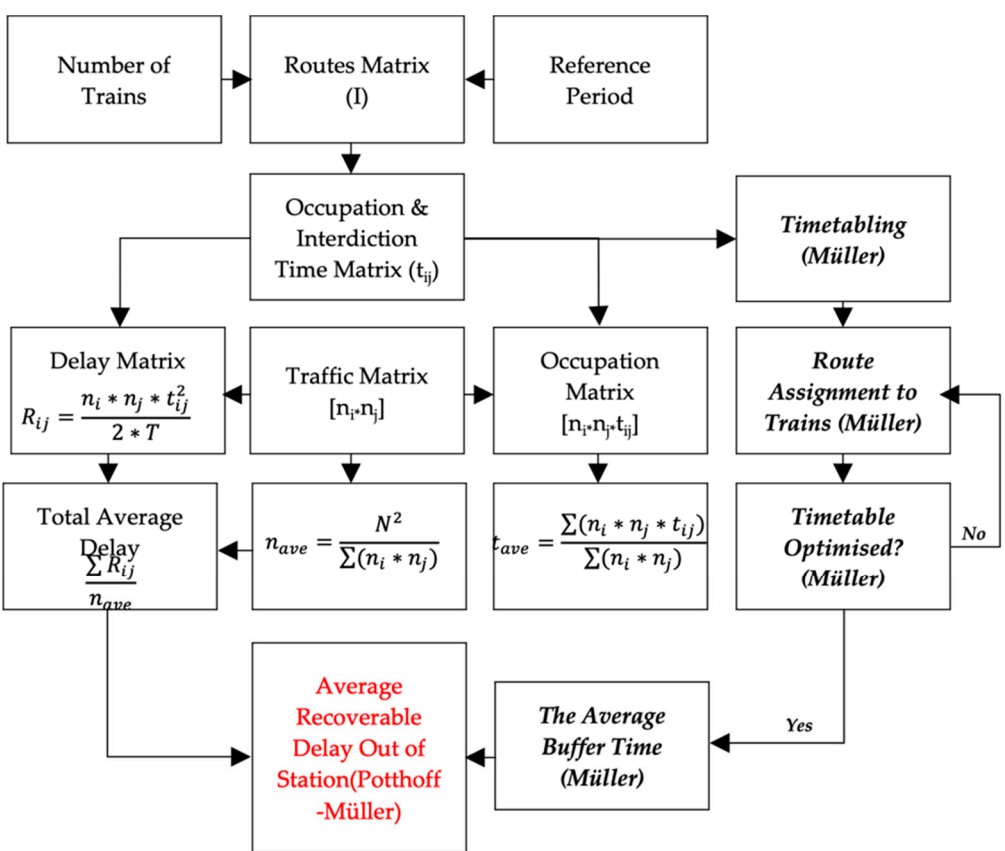

**Figure 3.** Potthoff and Müller methods integrated procedure.

### 3.2. Simulation Method

Simulation techniques are suitable to represent multiple assets of the railway infrastructure on a computer to avoid additional costs by providing an appropriate planning methodology largely supporting analysis and optimization of railway systems operation. OpenTrack® is a microscopic method specified for modeling dynamics of rolling stocks in interaction with infrastructure and timetable during the operation, by an iterative process capable to evaluate changes in each of these elements with system optimization purposes.

The simulation is based on user-defined input data: trains move on a defined track layout according to the timetable data. OpenTrack® uses a mixed discrete/continuous simulation process that calculates both the continuous numerical solution of the differential motion equations for the trains and the discrete processes of signal box states and delay distributions.

In this case, the average speed calculation is based on the official line's dossier of the network issued by the Infrastructure Manager RFI. The following assumptions are applied to the present research:

- First Train movement is modelled based on a mixed continuous-discrete method. The motion of trains in modelled by the solution of the differential motion equation (continuous) combined with signal information (discrete);
- Second Differential motion equation calculates the train's forward motion based on the maximum possible acceleration per time step;
- Second Train speed is obtained using integration and the distance covered using reintegration Figure 4;
- Second Type of occupation method is based on the default brake curves setting;
- Second Delay settings are the following: Small delays (minimum value for a train to be on delay, default = 1 s); Medium delays (default = 60 s); Large delays (default = 300 s).

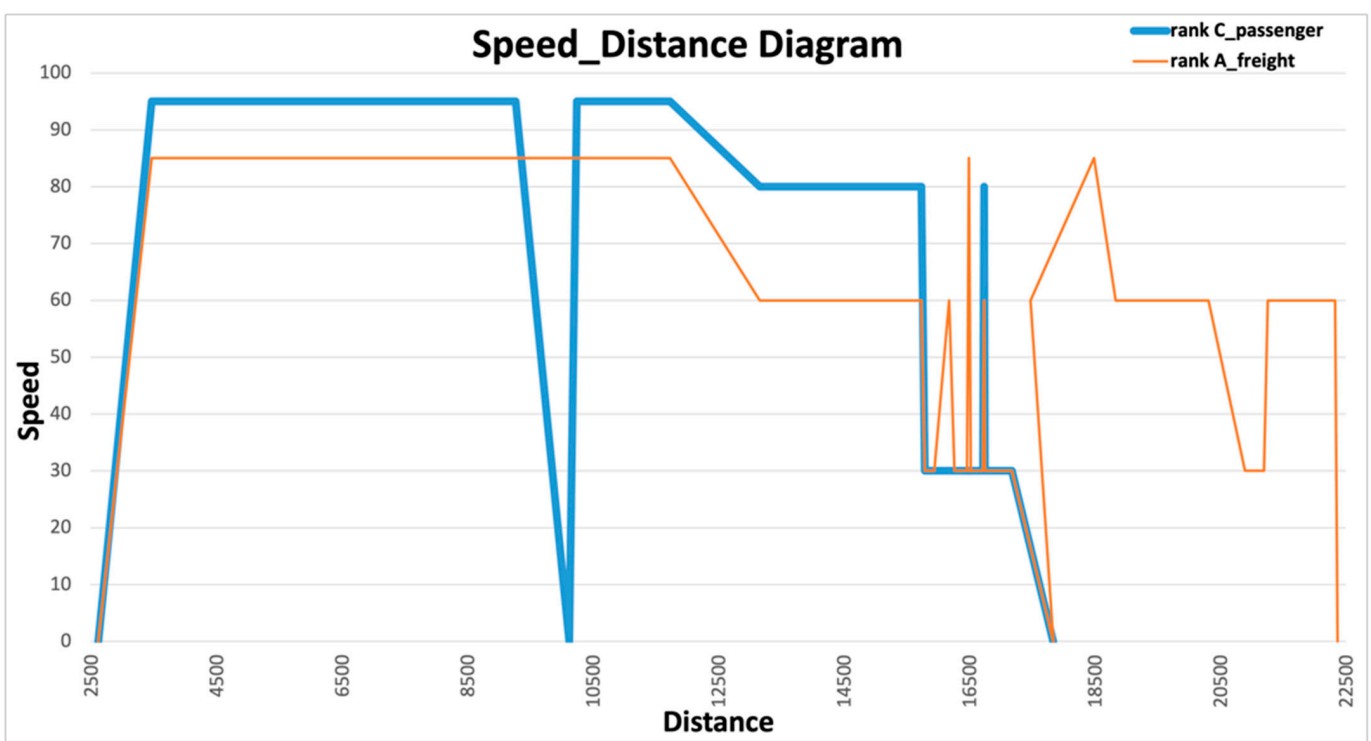

**Figure 4.** Speed–distance diagram for Trieste Centrale in blue, and Trieste Campo Marzio in red.

The speed calculations:

Euler's method was used to determine the speed at a time *t* in Equation (2):

$$(t) = v(t - \Delta t) + \Delta t \times \frac{dv}{dt}(t - \Delta t); v(t_0) = v0, \tag{2}$$

Using the motion equation, the calculation of the actual speed of a train is by integrating the formula below between the valid integration limits in Equation (3):

$$v = v0 + \int_{t1}^{t2} a.dt, \tag{3}$$

Similarly, the distance covered is by repeating the integration in Equation (4):

$$s = s0 + \int_{t1}^{t2} v.dt, \tag{4}$$

### 3.3. Comparative Analysis

The focus of the comparative analysis is to identify conflicts and delays as a difference between simulated and planned timetables, by identifying the potential bottlenecks and finally mitigating the conflicts and minimizing the delays. The stepwise structure proposed method is in Figure 5, (in brackets the sources of the results).

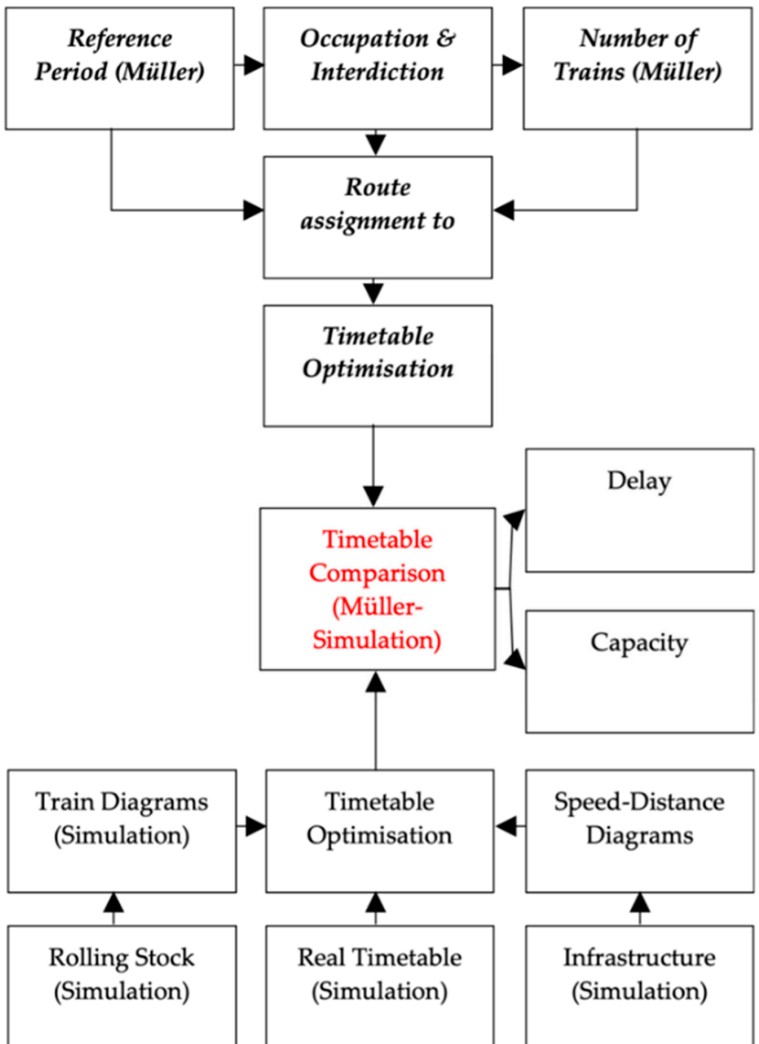

**Figure 5.** Müller and simulation methods integrated procedure.

## 4. Results

The outputs of the simulation allow for quantifying the punctuality ensured by the buffer times in the timetables, usable to recover arrival delays matured out of the stations (Trieste Centrale and Trieste Campo Marzio in the case study).

The results of timetable planning, in the present application for stations, are summarised in Figure 6, in terms of average delay indicators by day (18 h for passenger trains and 22 h for freight trains) and by a single train.

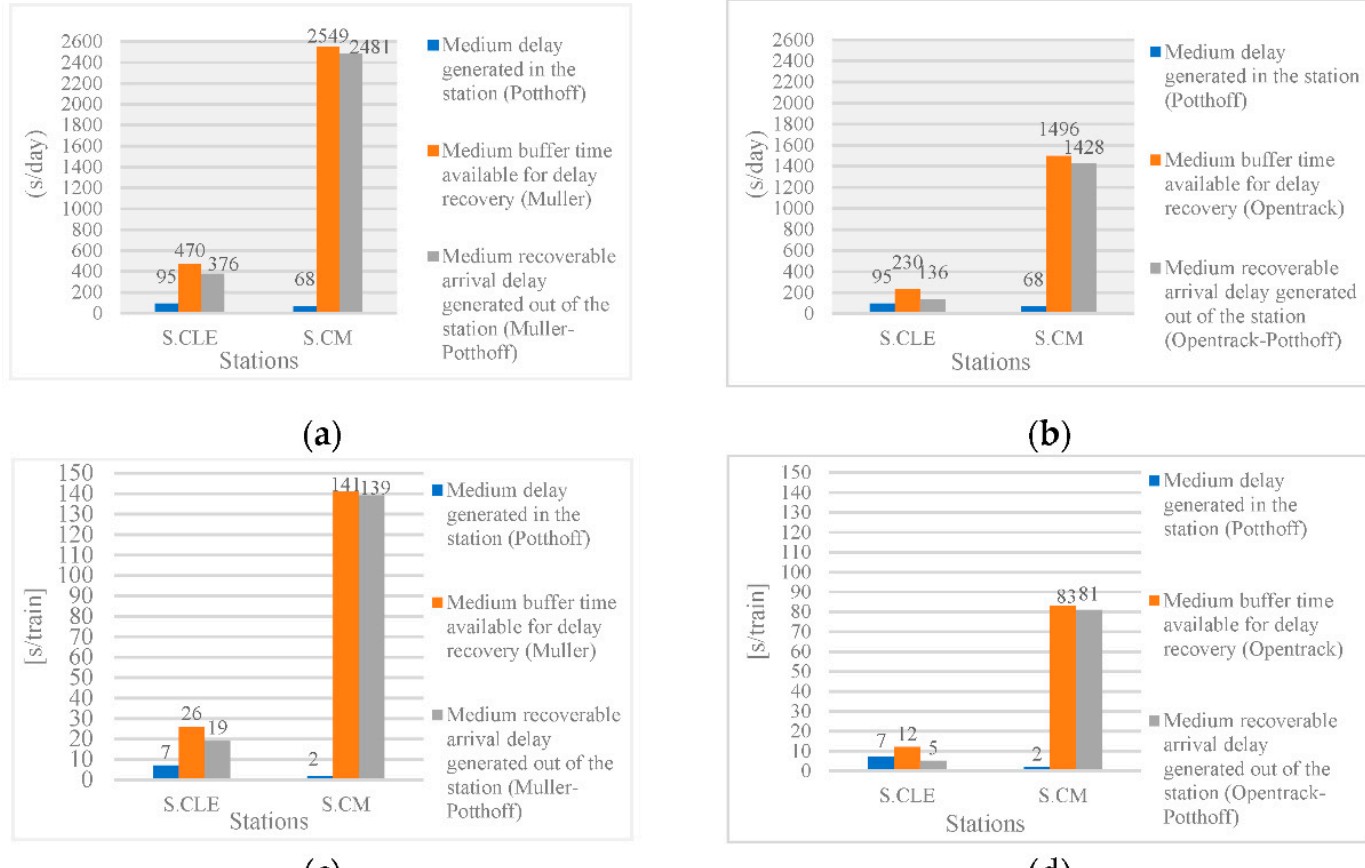

**Figure 6.** Delays, buffer times, and delay recovery potentials for Trieste Centrale and Trieste Campo Marzio stations by analytical and simulation methods listed as: (**a**) Result of Müller-Potthoff methods in seconds per day; (**b**) Result of Opentrack-Potthoff methods in seconds per day; (**c**) Result of Müller-Potthoff methods in seconds per train; (**d**) Result of Opentrack-Potthoff methods in seconds per train.

The comparisons between Potthoff and Müller analytical methods (on the left) and between Potthoff and OpenTrack® simulation (on the right) are readable in terms of average values for:

- Delay due to conflicts in the station (calculated by the Potthoff method);
- Total buffer time available, according to the planned timetable, respectively, resulting by Müller (on the left) and OpenTrack® simulation (on the right);
- Delay matured out of the station is recoverable by the residual buffer (difference between available buffer and delay matured in the station).

In Trieste Centrale, the total buffer is respectively 26 s/train according to the Müller method and much smaller (12 s/train) according to the OpenTrack® simulation. This is a relevant difference, which highlights the comparative level of confidence and the corresponding presumed application fields of the concerned literature methods, as discussed below.

The potential for recovering delays generated out of the station is correspondingly estimable in 21 and 7 s/train, values that highlight a quite low residual buffer due to a high density of traffic.

In terms of methodologies, the simulation provides results that are more prudential, both in Trieste Centrale and in Trieste Campo Marzio, where the traffic density is anyway quite lower.

This is mainly due to the network effect, which is typically included in the simulation; meanwhile, the analytical methods (Müller as well as Potthoff) are not considering any interaction between the stations and the surrounding network.

## 5. Discussion

The obtained results show that the analytical methods based on the timetable, such as that proposed by Müller, provide higher capacity values, for both passenger and freight stations. Meanwhile, the simulations by OpenTrack® are considering the network effects, which result in longer interdiction times and finally lower capacity.

These are very preliminary results, which provide interesting hints and exemplifications of combined line-station effects. Further validation and generalization activities on methodological approaches performances will include:

- Quantification of effects of timetable compression (e.g., by the UIC 406 method) on capacity and punctuality of both stations and lines;
- Further tests based on various traffic simulation tools at the network level;
- The sensitivity of the network capacity to upgrades of the signalling systems.

## 6. Conclusions

The final analyses demonstrate that the deviations between planned and simulated timetables are significant, and the evaluation of the capacity obtained by the analytical (Müller) method is prudential and the intrinsic reserve of punctuality set up by this method is effectively usable for the increase of capacity by an optimized timetabling process.

In this respect, we can presently conclude that:

- The use of the analytical method appears in line with the scope of high-level analyses for the identification of the most appropriate infrastructure layouts and signalling systems to adopt in a long-term perspective, independently upon a specific timetable structure;
- The implementation of a simulation model is necessary for in-depth analyses aiming at the optimization of the use of capacity and the timetable structure itself.

Nevertheless, further developments of the research should include a larger testing phase to check systematically the sensibility of the results to various timetables structure and to different signaling systems, e.g., according to the various ERTMS levels.

Furthermore, critical future research needs to be addressing the area related to the applicability of Internet of Things technologies for sustainable railway transportation [56].

**Author Contributions:** The distribution of the contributions and the corresponding authorship of the two authors (A.K. and S.R.) in the various sections of the paper is equal. All authors have read and agreed to the published version of the manuscript.

**Funding:** This research received no external funding.

**Institutional Review Board Statement:** Not applicable.

**Informed Consent Statement:** Not applicable.

**Data Availability Statement:** Data sharing not applicable. No new data were created or analyzed in this study. Data sharing is not applicable to this article.

**Conflicts of Interest:** The authors declare no conflict of interest.

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
