# Peer review of "Capacity Assessment in Freight-Passengers Complex Railway Nodes: Trieste Case Study"

_infrastructures, doi:10.3390/infrastructures7080106_

Round 1
Reviewer 1 Report
Review report on the manuscript “infrastructures-1858513”
Capacity Assessment in Freight-Passengers Complex Railway 2 Nodes: Trieste Case Study
Atieh Kianinejadoshah, Stefano Ricci
First, the calculation relationships applied in the Potthoff and Muller methods are missing, so it is difficult to verify the results in Figure 5. Also, in the case study analyzed, input data used, such as the number of trains and their type are missing.
The study shows that in Trieste Centrale station, the total buffer is 26 s/train, according to the Muller method, and 12 s/train according to the OpenTrack simulation. Considering that both results are predictive, it is not clear what role their comparison can have in practice.
Marginal observations refer to the Figure 2, where the colors of the graphs are not correlated and to the Figure 3, where the units of measure are missing on the graph and the word "rango" should be in English.
Author Response
Dear Reviewer
Thank you for the valuable suggestions on the paper " Capacity Assessment in Freight-Passengers Complex Railway Nodes: Trieste Case Study".
We have included the comments and responded to them individually in the uploaded file, indicating exactly how we addressed each concern, "please see the attachment."
We hope the revised manuscript will better suit the Infrastructure Journal and are happy to consider further revisions, thank you for your continued interest in our research.
Sincerely,
Atieh Kiani

Reviewer 2 Report
The paper contains a simple but complex presentation of a selected problem concerning capacity assessment and timetabling problematics. The topic is worth publishing, however, the manuscript should be improved according to two main aspects.
First, please enlarge the literature review. Now, nine of the 13 publications [7-19] commented in section 2.1 have authorship of one or both Authors of reviewed manuscript. The citation of your own publications probably has a sense here, but the broader literature review is necessary.
For example, I found many search results in MDPI publications.
Using the keyword “timetabling”: in all journals 133 positions, 29 from the Year 2022, 1 position from the journal “Infrastructures” (Exploring the Applicability of Location-Based Services to Delineate the State Public Transport Routes Integratedness within the City of Johannesburg, august 2018).
Using the keyword “railway node”: in all journals 37 positions, 7 from the Year 2022.
I show other exemplary and interesting papers (potential valuable to cite):
Oneto, L., Fumeo, E., Clerico, G., ,Canepa, R., Papa, F., Dambra, C., Mazzino, N., Anguita, D., 2017. Dynamic Delay Predictions for Large-Scale Railway Networks: Deep and Shallow Extreme Learning Machines Tuned via Thresholdout. IEEE Transactions on Systems, Man, and Cybernetics: Systems. 47 (10), 2754-2767.
https://doi.org/10.1109/TSMC.2017.2693209
Zhang, T., Li, D., Qiao, Y., 2018. Comprehensive optimization of urban rail transit timetable by minimizing total travel times under time-dependent passenger demand and congested conditions. Appl. Math. Model. 58, 421–446.
https://doi.org/10.1016/j.apm.2018.02.013
Lianhua, T., Xingfang, X., 2022. Optimization for operation scheme of express and local trains in suburban rail transit lines based on station classification and bi-level programming. Journal of Rail Transport Planning & Management. 21,100283.
https://doi.org/10.1016/j.jrtpm.2021.100283
Second, add the "conclusions" (as an independent section) with general commentary to the presented results according to the goals presented in the first part of the paper and the enhanced literature review.
Author Response

(The authors gave the same response as above.)

Round 2
Reviewer 1 Report
The authors made additions to the manuscript, in accordance with the reviewers' recommendations. The methodology, bibliography and conclusions were revised, and thus the work was significantly improved. In my opinion, the manuscript can be considered for publication in its current form.